# PREDICTION-CONSISTENT KOOPMAN AUTOENCODERS

## ABSTRACT

Data-driven modeling of high-dimensional spatio-temporal dynamical systems, which are often governed by nonlinear partial differential equations (PDEs), poses a serious challenge in the absence of sufficient or high-quality training data. Recently developed Koopman autoencoders (KAEs) leverage the expressivity of deep neural networks (DNNs) and the spectral structure of Koopman operator to learn a reduced-order feature space exhibiting simpler linear dynamics. However, limited and noisy training datasets present a significant roadblock and results in a lack of generalizability due to inconsistency in training data. In this paper we propose the prediction-consistent Koopman autoencoder (pcKAE) which is capable of generating accurate long-term predictions even with limited and noisy training data. We introduce a consistency regularization term that enforces consistency among predictions at different time-steps, making pcKAE more robust and generalizable compared to its counterparts. An analytical justification is presented for such consistency regularization using the Koopman spectral theory. Experimentally, we demonstrate that with limited training data, pcKAE outperforms existing state-of-the-art KAE models for several test-cases, ranging from simple pendulum to kinetic plasmas, fluid flows and sea surface temperature data.

## 1 INTRODUCTION

Long-tern forecasting sequential time-series data generated from a nonlinear dynamical system is a central problem in engineering. For classification or prediction of such time-series data, a precise analysis of the underlying dynamical system provides valuable insights to generate appropriate features and interpretability to any computational algorithm. This gives rise to a family of physics constrained learning (PCL) algorithms that incorporate constraints arising from physical consistency of the governing dynamical system.

Recently, a physics constrained approach for time-series learning based on Koopman methods (Rowley et al., 2009) has been introduced. This approach uses an infinite-dimensional linear operator that encodes a nonlinear dynamics completely. The Koopman operator (Koopman, 1931) provides a computationally preferable linear method to analyze and forecast dynamical systems. However, Koopman operator maps between infinite-dimensional function spaces, and hence, cannot be *in general* computationally represented without a finite-dimensional projection . Machine learning techniques utilize a finite-dimensional approximation of the Koopman operator by assuming the existence of a finite-dimensional Koopman invariant function space. This is usually achieved by an autoencoder network that maps the state-space into a latent space where the dynamics can be linearly approximated by the finite-dimensional Koopman operator encoded via a linear layer. This Koopman autoencoder (KAE) approach is attractive as it strikes a balance between the expressivity of deep neural networks and interpretability of PCL. Also, Koopman modal approximation can be readily used for stability analysis (Mauroy & Mezić, 2016) and control (Peitz et al., 2020; Goswami & Paley, 2022) of the underlying dynamical system in a data-driven fashion. Recent literature (Azencot et al., 2020) also investigates the existence of a backward Koopman operator in order to impose an extra consistency constraint on the latent space linear map, giving rise to consistent Koopman autoencoder (cKAE) algorithm.

This manuscript proposes a new algorithm for consistent long-term prediction of time-series data generated from a nonlinear (possibly high-dimensional) dynamical system using a prediction consistency constraint with Koopman autoencoder framework. This algorithm, dubbed as prediction consistent Koopman autoencoder (pcKAE), compares the predictions from different initial time-instances to a

final time in the *latent space*. This is in contrast with prior KAE methodologies, where the enforcement of multi-step look-ahead prediction loss is reliant on *labelled* data (Azencot et al., 2020; Lusch et al., 2018; Takeishi et al., 2017), a constraint when such data is scarce. The pcKAE is different since it evaluates predictions relative to each other, bypassing direct comparison with *labelled* data. This alleviates limitations on the maximum look-ahead step imposed by a limited training dataset. The pcKAE algorithm also differs from cKAE (Azencot et al., 2020) in checking consistency because, unlike the latter, it does not require the existence of a backward dynamics. It is analytically shown that a KAE spanning a Koopman invariant subspace must satisfy the prediction consistency constraint and by enforcing it, pcKAE leads to higher expressivity and generalizability. This enforcement of prediction consistency in effect reduces the sensitivity of the KAE to noise, decreases the variance or uncertainty in future predictions and make it more robust with less training data as shown in the experiments. Consistency regularization have been used for semi-supervised learning in classification problems (Sajjadi et al., 2016; Englesson & Azizpour, 2021; Fan et al., 2023), but to the best of our knowledge, this is the first attempt to regularize multi-step prediction consistency for dynamical system learning.

## 2 BACKGROUND

### 2.1 RELATED WORK

**Recurrent neural networks:** The idea of forecasting dynamical systems using neural networks, particularly using recurrent neural networks (RNNs), dates back to several decades (Bailer-Jones et al., 1998; Aussem, 1999). Recent advances in big data, machine learning algorithms and computational hardware, have rekindled the interest in this domain. Emergence of innovative architectures such as Long Short-Term Memmory (LSTM) (Hochreiter & Schmidhuber, 1997), or gated neural networks (GRUs) (Chung et al., 2014) have enhanced the prediction capabilities of NN models significantly. However, training an RNN is fraught with vanishing and exploding gradient problems (Bengio et al., 1994) which can be solved by approaches like analyzing stability (Miller & Hardt, 2019) or using unitary hidden weight matrices (Arjovsky et al., 2016). But these can affect the short term modeling capability by reducing the expressivity of the RNNs (Kerg et al., 2019a). Another drawback of RNNs is the lack of generalizability and interpretability especially when used for predicting a physical system. To make RNN physically consistent, a number of methods are presented to take physical constraints into account. These range from making a physics guided architecture (Jia et al.), relating it to dynamical systems (Sussillo & Barak, 2013) or differential equations (Chang et al., 2019). Hamiltonian neural networks (Greydanus et al., 2019) aims to learn the conserved Hamiltonian as a physical constraint but works only for lossless systems. In this paper, we use the linear operator-based recurrent archiecture along with a physical prediction constraint to develop a more generalizable prediction model.

**Koopman-based methods:** The Koopman operator-based approach exploits the linearity (Rowley et al., 2009) in an infinite dimensional function space to yield a linearly recurrent prediction scheme. A number of non-neural dictionary-based approach relying on the dynamic mode decomposition (DMD) (Schmid, 2010) are proposed (Williams et al., 2015; 2016; Peitz et al., 2020; Goswami & Paley, 2017; 2022). However, these methods implicitly assumes a dictionary spanning a Koopman invariant subspace. Neural approaches using Koopman autoencoder (KAE) provides a better alternative to define a Koopman invariant function space where the dynamics can be linearly approximated (Takeishi et al., 2017; Otto & Rowley, 2019; Lusch et al., 2018). (Lange et al., 2021) showcased the superiority of Koopman-based spectral methods over LSTM, Gated Recurrent Unit (GRU), and Echo State Networks (ESNs) due its ability to extract "slow" frequencies which can have significant effect for the long-term predictions. Furthermore, the autoencoder architecture is particularly well-suited (order reduction) for modeling high-dimensional physical systems, the primary goal of this work. However, most of the KAE models just focus on multi-step prediction errors and without any test on consistency of predictions. Recent work (Azencot et al., 2020) on backward Koopman operator provides a consistency test by making the predictions forward and backward consistent, but works only when a backward dynamics is well-defined.

## 2.2 KOOPMAN THEORY: AN OVERVIEW

Consider a discrete-time dynamical system on a $N_d$-dimensional compact manifold $\mathcal{M}$, evolving accoring to the flowmap $\mathbf{f} : \mathcal{M} \mapsto \mathcal{M}$:

$$\mathbf{x}_{n+1} = \mathbf{f}(\mathbf{x}_n), \quad \mathbf{x}_n \in \mathcal{M}, \quad n \in \mathbb{N} \cup \{0\}. \tag{1}$$

Let $\mathcal{F}$ be a Banach space of complex-valued observables $\psi : \mathcal{M} \to \mathbb{C}$. The discrete-time *Koopman operator* $\mathcal{K} : \mathcal{F} \to \mathcal{F}$ is defined as

$$\mathcal{K}\psi(\cdot) = \psi \circ \mathbf{f}(\cdot), \quad \text{with } \psi(\mathbf{x}_{n+1}) = \mathcal{K}\psi(\mathbf{x}_n) \tag{2}$$

where $\mathcal{K}$ is infinite-dimensional, and linear over its argument. The scalar observables $\psi$ are referred to as the Koopman observables. Koopman eigenfunctions $\phi$ are special set of Koopman observables that satisfy $(\mathcal{K}\phi)(\cdot) = \lambda\phi(\cdot)$, with an eigenvalue $\lambda$. Considering the Koopman eigenfunctions span the Koopman observables, a vector valued observable $\mathbf{g} \in \mathcal{F}^p = [\psi_1\ \psi_2\ \ldots\ \psi_p]^{\mathrm{T}}$ can be expressed as a sum of Koopman eigenfunctions $\mathbf{g}(\cdot) = \sum_{i=1}^{\infty} \phi_i(\cdot)\mathbf{v}_i^{\mathbf{g}}$, where $\mathbf{v}_i^{\mathbf{g}} \in \mathbb{R}^p, i = 1, 2, \ldots$, are called the *Koopman modes* of the observable $\mathbf{g}(\cdot)$. This modal decomposition provides the growth/decay rate $|\lambda_i|$ and frequency $\angle\lambda_i$ of different Koopman modes via its time evolution

$$\mathbf{g}(\mathbf{x}_t) = \sum_{i=1}^{\infty} \lambda_i^t \phi_i(\mathbf{x}_0)\mathbf{v}_i^{\mathbf{g}}. \tag{3}$$

The Koopman eigenvalues and eigenfunctions are properties of the dynamics only, whereas the Koopman modes depend on the observable.

Koopman modes can be analyzed to understand the dominant characteristics of a complex dynamical system and getting traction in fluid mechanics (Rowley et al., 2009), plasma dynamics (Nayak et al., 2021), control systems (Peitz et al., 2020), unmanned aircraft systems (Narayanan et al., 2023), and traffic prediction (Avila & Mezić, 2020). In addition, it is also being used for machine learning tasks and training deep neural networks (Dogra & Redman, 2020). Several methods have also been developed to compute the Koopman modal decomposition, e.g., DMD and EDMD (Schmid, 2010; Williams et al., 2015), Ulam-Galerkin methods, and deep neural networks (Otto & Rowley, 2019; Yeung et al., 2019). In this paper, we primarily focus on long-term prediction of autonomous dynamical systems using Koopman modes with autoencoder networks.

## 3 METHOD

### 3.1 PREDICTION VIA KOOPMAN INVARIANT SUBSPACE: KOOPMAN AUTOENCODERS (KAE)

Koopman operator, being an infinite-dimensional one, must be projected onto a finite-dimensional basis for any practical prediction algorithm. The equation in (3) can be truncated to finite set of terms in the presence of a finite-dimensional Koopman invariant subspace. However, discovering such finite-dimensional Koopman invariant subspaces purely from data is a challenging task, and an active area of research. Recent works (Takeishi et al., 2017; Lusch et al., 2018; Otto & Rowley, 2019) leverage the nonlinear function approximation capabilities of neural networks to find a suitable transformation from the state space $\mathcal{M}$ to a Koopman invariant subspace via a latent state ($\mathbf{z}$). These models are encompassed within the broader umbrella of Koopman autoencoders (KAE) which strives to find a suitable transformation from state space to the Koopman observable space where the dynamics is linear, and can be easily learned. The basic operation of KAE can be decomposed into three components (4), *i)* encoding $(\mathbf{\Psi}_e(\cdot)) : \mathcal{M} \to \mathbb{R}^{N_l}$ that transforms the original state $\mathbf{x}$ to a Koopman observable $\mathbf{z} \in \mathbb{R}^{N_l}$, *ii)* advancing the dynamics in that transformed space through a linear operator $(K \in \mathbb{R}^{N_l \times N_l})$, and *iii)* decoding $(\mathbf{\Psi}_d(\cdot))$ back to the original state space:

$$\mathbf{z}_n \approx \mathbf{\Psi}_e(\mathbf{x}_n) \quad \to \quad \mathbf{z}_{n+k} = K^k \cdot \mathbf{z}_n \quad \to \quad \mathbf{x}_{n+k} \approx \mathbf{\Psi}_d(\mathbf{z}_{n+k}) = \hat{\mathbf{x}}_{n+k}, \tag{4}$$

where $K$ is the finite-dimensional restriction of $\mathcal{K}$, which advances $\mathbf{z}$ by $k$ time-steps (or time samples), and $\hat{\mathbf{x}}$ denotes the KAE reconstruction of the state. Note that for traditional autoencoders the encoding operation typically result in dimensionality reduction which is beneficial when dealing with high-dimensional dynamical systems. Traditionally, since we are mostly interested in future prediction of the state, KAE is trained by minimizing the difference between $\hat{\mathbf{x}}_{n+k}$ and $\mathbf{x}_{n+k}$, with

$k \in \mathbb{Z}_+ \cup \{0\}$. The loss functions traditionally used for training KAEs are the identity loss $\mathcal{L}_{\text{id}}$ ($k = 0$), and the forward loss $L_{\text{fwd}}$ ($k = 1, 2, \ldots$) (Azencot et al., 2020).

$$\mathcal{L}_{id} = \frac{1}{2M}\sum_{n=1}^{M}||\hat{\mathbf{x}}_n - \mathbf{x}_n||_2, \quad \mathcal{L}_{fwd} = \frac{1}{2k_m M}\sum_{k=1}^{k_m}\sum_{n=1}^{M}||\hat{\mathbf{x}}_{n+k} - \mathbf{x}_{n+k}||_2^2, \tag{5}$$

where $|| \cdot ||_2$ is the 2-norm, $M$ is the number of samples over which we want to enforce the loss, and $k_m$ is the maximum value of $k$, i.e. the maximum look-ahead step for multi-step training. Recent work in (Azencot et al., 2020) demonstrated that including backward dynamics ($K_b \in \mathbb{R}^{N_l \times N_l}$) leads to better stability resulting in more accurate long-term predictions. The idea is to incorporate a backward loss term ($\mathcal{L}_{\text{bwd}}$) with forward-backward consistency loss ($\mathcal{L}_{\text{con}}$),

$$\mathcal{L}_{bwd} = \frac{1}{2k_m M}\sum_{k=1}^{k_m}\sum_{n=1}^{l}||\hat{\mathbf{x}}_{n-k} - \mathbf{x}_{n-k}||_2^2 \tag{6}$$

$$\mathcal{L}_{con} = \sum_{i=1}^{N_b}\frac{1}{2i}||K_{bi\star}K_{\star i} - I_{N_l}||_F + \frac{1}{2i}||K_{\star i}K_{bi\star} - I_{N_l}||_F, \tag{7}$$

where $N_l$ is the latent space dimension, $K_{bi\star}$ is the upper $i$ rows of $K_b$, $K_{\star i}$ is the $i$ left most columns of $K$, $|| \cdot ||_F$ denotes the Frobenius norm, and $I_{N_l} \in \mathbb{R}^{N_l \times N_l}$ is the identity matrix of dimension $N_l \times N_l$.

## 3.2 PREDICTION-CONSISTENT KOOPMAN AUTOENCODER (PCKAE)

In this section, we describe our proposed prediction-consistent Koopman autoencoder (pcKAE) architecture. The key idea is to introduce a consistency regularization term for training KAE in order to enforce consistency among future predictions. The fundamntal idea derives from the time-invariance property of the autonomous dynamical systems. In the context of (1), the prediction consistency can be stated as the direct result of the following:

$$\mathbf{f}^\kappa(\mathbf{x}_n) = \mathbf{f}^{\kappa+k}(\mathbf{x}_{n-k}), \quad \forall n, \kappa \in \mathbb{Z}_+ \cup \{0\}, \ k = 1, 2, \ldots, n, \tag{8}$$

where $\mathbf{f}^\kappa = \mathbf{f} \circ \mathbf{f} \circ \ldots (\kappa \text{ times}) \ldots \circ \mathbf{f}$, with $\circ$ being the composition operator. One key aspect of our approach is that instead of enforcing this consistency in the original state space, we do so in the Koopman invariant subspace (latent space). This helps avoid computation in the high-dimensional space, as well as provide robustness to noise. We justify enforcing such consistency in the latent space by summarizing the desired property of a consistent autoencoder $\mathbf{\Psi}_e$ and constructing the architecture of this framework. The autoencoder $\mathbf{\Psi}_e(\cdot) = [\psi_1(\cdot), \ldots, \psi_{N_l}(\cdot)]$ defines a vector valued observable of the state space $\mathcal{M}$ where each $\psi_i$ is a scalar valued function. Let $\mathcal{G}$ be the span of $\{\psi_1, \ldots, \psi_l\}$. In order for the linear recurrent matrix $K$ to asymptotically approximate the projection of Koopman operator $\mathcal{K}$ on $\mathcal{G}$, the latter needs to be Koopman invariant, i.e., $\mathcal{K}\mathcal{G} \subset \mathcal{G}$. The Koopman invariance will ensure the linear recurrent structure

$$\hat{\mathbf{x}}_{n+1} = \mathbf{\Psi}_d \circ K \circ \mathbf{\Psi}_e(\mathbf{x}_n). \tag{9}$$

The proposed architecture introduces a multi-step prediction consistency loss (not against labelled data) in the latent space in order to strengthen the Koopman invariance of the latent function space $\mathcal{G}$. In the following, we summarize the desirable prediction consistency of the encoder functions upon which the proposed method is constructed.

**Theorem 1.** *Let $\mathbf{\Psi}_e(\cdot) = [\psi_1(\cdot), \ldots, \psi_{N_l}(\cdot)]^T \in \mathcal{F}^{N_l}$ denotes a vector valued function comprised with scalar functions $\psi_i(\cdot) \in \mathcal{F}$. The latent function space $\mathcal{G} = \{\psi_1(\cdot) \ldots, \psi_{N_l}(\cdot)\}$ forms a Koopman invariant subspace with respect to the system dynamics (1) if and only if there exists $K \in \mathbb{R}^{N_l \times N_l}$ such that*

$$\mathbf{\Psi}_e(\mathbf{x}_{n+\kappa}) = K^\kappa \mathbf{\Psi}_e(\mathbf{x}_n) \tag{10}$$

*for all $n \geq 0$ and $\kappa \geq 1$.*

*Proof.* ($\Rightarrow$) Suppose $\mathbf{\Psi}_e(\cdot)$ is such defined that span($\mathcal{G}$) is a Koopman invariant subspace. We prove this part via induction. For any $n \geq 0$, (10) is trivially satisfied for $\kappa = 1$ with some matrix $K$. Now suppose it is satisfied for a fixed $\kappa \in \mathbb{N}$. Then, $\mathbf{\Psi}_e(\mathbf{x}_{n+\kappa+1}) = \mathbf{\Psi}_e \circ \mathbf{f}(\mathbf{x}_{n+\kappa}) = \mathcal{K}\mathbf{\Psi}_e(\mathbf{x}_{n+\kappa}) =$

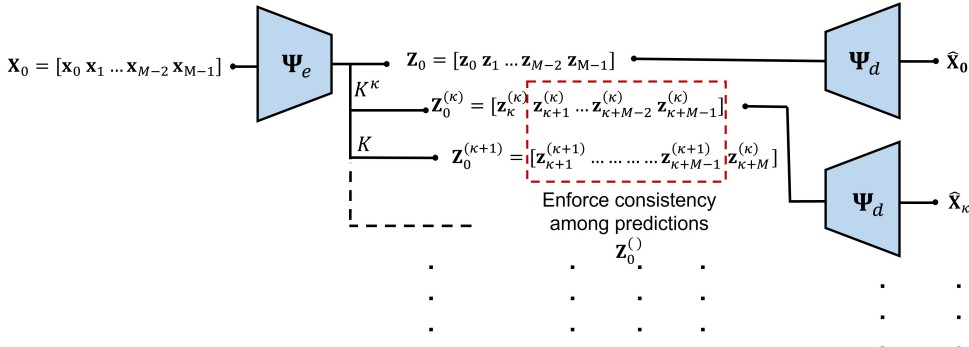

Figure 1: Illustration of enforcement of prediction consistency among predictions in Koopman invariant latent space.

$\mathcal{K}K^{\kappa}\mathbf{\Psi}_e(\mathbf{x}_n) = K^{\kappa}\mathcal{K}\mathbf{\Psi}_e(\mathbf{x}_n) = K^{\kappa}\mathbf{\Psi}_e \circ \mathbf{f}(\mathbf{x}_n) = K^{\kappa}\mathbf{\Psi}_e(\mathbf{x}_{n+1}) = K^{\kappa+1}\mathbf{\Psi}_e(\mathbf{x}_n)$, where the first two steps are by the definition of Koopman operator, next two steps are by the linearity of the same, and the last step is by the induction anchor.

($\Leftarrow$) Suppose $\exists\, K \in \mathbb{R}^{N_l \times N_l}$ such that (10) is satisfied for all $n \geq 0$ and $\kappa \geq 1$. Then it must be satisfied for $n = 0$ and $\kappa = 1$, i.e., $\mathbf{\Psi}_e(\mathbf{x}_1) = K\mathbf{\Psi}(\mathbf{x}_0)$. Since $\mathbf{x}_0$ can be anything in $\mathcal{M}$, this yields $\mathbf{\Psi}_e \circ \mathbf{f}(\mathbf{x}) = K\mathbf{\Psi}_e(\mathbf{x})$ for all $\mathbf{x} \in \mathcal{M}$, i.e., $\mathbf{\Psi}_e \circ \mathbf{f}(\cdot) = K\mathbf{\Psi}_e(\cdot)$. Let $g \in \text{span}(\mathcal{G})$ be an observable. Then, $\exists\, \alpha \in \mathbb{R}^l$ such that $g(\cdot) = \alpha^T \mathbf{\Psi}_e(\cdot)$. Now, $\mathcal{K}g(\cdot) = g \circ \mathbf{f}(\cdot) = \alpha^T \mathbf{\Psi}_e \circ \mathbf{f}(\cdot) = \alpha^T K\mathbf{\Psi}_e(\cdot)$, i.e., $\mathcal{K}g \in \text{span}(\mathcal{G})$. Hence, $\text{span}(\mathcal{G})$ forms a Koopman invariant subspace. $\square$

Theorem 1 ensures that, in the latent space, encoded state at any time instant must be equal to the encoded state of any previous time multiplied with $K^{\kappa}$ where $\kappa$ denotes the time-step between those two instances. It also forces the latent state to have linear time invariant dynamics with a constant state-transition matrix $K$. Theorem 1 provides a method to ensure the autoencoder span a Koopman invariant subspace by enforcing prediction consistency as follows.

**Corollary 1.** *For a KAE $\mathbf{\Psi}_e : \mathcal{M} \to \mathbb{R}^{N_l}$, if the associated function space $\mathcal{G}$ is Koopman invariant, then for any $n \geq 0$ and $\kappa \geq 1$, $K^{\kappa-p}\mathbf{\Psi}_e(\mathbf{x}_{n+p}) = K^{\kappa-q}\mathbf{\Psi}_e(\mathbf{x}_{n+q})$ for all $p, q = 1, \ldots, \kappa - 1$, i.e.,*

$$K^{\kappa-1}\mathbf{\Psi}_e(\mathbf{x}_{n+1}) = K^{\kappa-2}\mathbf{\Psi}_e(\mathbf{x}_{n+2}) = \ldots (p \neq q)$$

Corollary 1 is essentially heart of the prediction consistency loss. In order to define our loss function, let us consider the reference time-step to be $n$. For a batch size of $M$ (in latent space $\mathbf{Z}_n = [\mathbf{z}_n\, \mathbf{z}_{n+1} \ldots \mathbf{z}_{n+M-1}]$), let the KAE advance the dynamics up to $\kappa_m$ time-steps, giving rise to a set of batches $\mathbf{Z}_n^{(\kappa)} = [\mathbf{z}_{n+\kappa}^{(\kappa)}\, \mathbf{z}_{n+1+\kappa}^{(\kappa)} \ldots \mathbf{z}_{n+M+\kappa-1}^{(\kappa)}]$ with $\kappa = 1, 2, \ldots, \kappa_m$, where the superscript $(\kappa)$ denotes the number of operations of $K$. The consistency loss $\mathcal{L}_{\text{pc}}$ (Fig. 1) can be defined as,

$$\mathcal{L}_{pc} = \frac{1}{2\kappa_m} \sum_{\kappa=1}^{\kappa_m} \mathcal{L}_{\kappa}, \quad \mathcal{L}_{\kappa} = \frac{1}{(M-1)} \sum_{p=1}^{M-1} \frac{1}{p} \sum_{q=1}^{p} ||\mathbf{z}_{n+\kappa+p}^{(\kappa)} - \mathbf{z}_{n+\kappa+p}^{(\kappa+q)}||_2^2. \tag{11}$$

The total loss ($\mathcal{L}_{\text{tot}}$) for training pcKAE is given by,

$$\mathcal{L}_{\text{tot}} = \gamma_{\text{id}}\mathcal{L}_{\text{id}} + \gamma_{\text{fwd}}\mathcal{L}_{\text{fwd}} + \gamma_{\text{bwd}}\mathcal{L}_{\text{bwd}} + \gamma_{\text{con}}\mathcal{L}_{\text{con}} + \gamma_{\text{pc}}\mathcal{L}_{\text{pc}}, \tag{12}$$

Where $\gamma_{\text{id}}, \gamma_{\text{fwd}}, \gamma_{\text{bwd}}, \gamma_{\text{con}}$ and $\gamma_{\text{pc}}$ are the corresponding weights.

**Remark 1.** *The prediction consistency loss enforces a Koopman invariant subspace by virtue of the time-homogeneity of the latent-state dynamics, i.e., the Koopman operator is time-invariant for an autonomous system. This provides a* physics constrained learning *method for dynamical system prediction even from a smaller dataset.*

### 3.3 BASELINE

The performance of the proposed prediction-consistent Koopman autoencoder (pcKAE) is compared against the state-of-the-art KAE models forecasting models for physical systems. As discussed earlier, several neural-network based models such as recurrent neural networks (RNNs) (Chung et al., 2014; Kerg et al., 2019b; Chang et al., 2019), Hamiltonian neural networks (HNNs) (Greydanus et al., 2019) have already been applied successfully in forecasting dynamical systems. However, these models do not fare well for very long-term predictions (Lange et al., 2021). Koopman-based autoencoder models (Lusch et al., 2018; Otto & Rowley, 2019) have been shown to be effective in capturing this long-term behavior, especially for high-dimensional systems. However, recently proposed consistent Koopman autoencoder (Azencot et al., 2020) which we will refer to as cKAE, has been shown to outperform all the mentioned methods for long-term predictions in terms of stability. We will consider cKAE as the baseline for benchmarking our proposed pcKAE for several datasets. We will also compare our results with dynamic autoencoders (DAEs) term used in (Azencot et al., 2020) to describe (Lusch et al., 2018; Otto & Rowley, 2019) with $\gamma_{\text{bwd}} = \gamma_{\text{con}} = 0$.

### 3.4 DATASETS

In order to streamline the discussion, we adopt the following notation: the complete dataset is denoted by $\mathbf{X} \in \mathbb{R}^{N_{\text{dim}} \times N_{\text{tot}}}$, where $N_{\text{dim}}$ is the dimension of the state-space, and $N_{\text{tot}}$ is the total number of time samples. We denote the training set as $\mathbf{X}_{\text{train}} \in \mathbb{R}^{N_{\text{dim}} \times N_{\text{train}}}$ and testing set as $\mathbf{X}_{\text{test}} \in \mathbb{R}^{N_{\text{dim}} \times N_{\text{test}}}$ with $N_{\text{train}}$ and $N_{\text{test}}$ being the number of time samples in training and testing set respectively.

#### 3.4.1 UNDAMPED PENDULUM:

The undamped pendulum is a classic textbook example of nonlinear dynamical system for benchmarking performance of data-driven predictive models (Azencot et al., 2020; Greydanus et al., 2019; Chen et al., 2020; Bounou et al., 2021; Azari & Erdogmus, 2022). Although, the pendulum dynamics do not fall under typical PDE-governed high-dimensional systems, it serves as an excellent illustrative example due to the wide range of physical systems they represent, ranging from orbital mechanics, micorwave cavities to select biological and quantum systems. We use the elliptical functions (Azencot et al., 2020; cKA) for exact solution of the motion governed by the ODE $\frac{d^2\theta}{dt^2} + \frac{g}{l}\sin\theta = 0$, where $\theta \in [0, 2\pi]$ denotes the angular displacement from equilibrium $\theta = 0$ in radians. The gravitational constant and length of the pendulum is denoted by $g$ and $l$ respectively. We consider $g = 9.8$ m/s$^2$, and $l = 1$ m with initial condition $\theta = \theta_0$ and $\frac{d\theta}{dt} = \dot\theta = 0$. We sample the solution at interval of 0.1 s upto 220 s, resulting in the dataset $\Theta \in \mathbb{R}^{2 \times 2200}$ with 2200 time samples. Note that the dimensionality of the state-space is two, since we have two states $\theta$ and $\dot\theta$. In order to mimic a high-dimensional system, we use a random orthogonal transformation (Azencot et al., 2020) to rotate it to higher dimensional ($N_{\text{dim}} = 64$) space, i.e. $\mathbf{X} = \mathbf{P} \cdot \Theta$, where $\mathbf{P} \in \mathbb{R}^{64 \times 2}$ is a random orthogonal matrix. The final dataset $\mathbf{X} \in \mathbb{R}^{64 \times 2200}$ consists of 2200 time samples of a state with dimension 64.

#### 3.4.2 OSCILLATING ELECTRON BEAM

Plasma systems are excellent candidates for data-driven modeling due to their high-diemnsionality, and inherent nonlinearity arising from the complex wave-particle interaction. A two dimensional electron beam oscillating under the influence of an external transverse magnetic flux is simulated using charge-conserving electromagnetic particle-in-cell (EMPIC) algorithm (Na et al., 2016). The solution domain (1 m × 1 m) is discrtized using unstructurd triangular mesh with 844 nodes, 2447 edges, and 1604 elements (triangles). The electric field data ($\mathbf{e} \in \mathbb{R}^{2447}$) is sampled at every 0.32 ns. Starting from 80 ns, total 1751 time samples are collected with the complete dataset $\mathbf{X} \in \mathbb{R}^{2447 \times 1751}$. More details regarding data generation are provided in the supplementary material.

#### 3.4.3 FLOW PAST CYLINDER

Flow past cylinder is another common dynamical system used for benchmarking data-driven models. The simulation was carried out using MATLAB's FEAtool. A cylinder with diameter 0.1 m is located at a height of 0.2 m. The fluid is charactized by its density $\rho = 1$ Kg/m$^3$, and dynamic viscosity $\mu = 0.001$ Kg/m s. The flow is unsteady with a maximum velocity of 1 m/s and mean velocity $\frac{2}{3}$

of the maximum velocity. The simulation is run for 80 s until the steady state is achieved, and the horizontal $u$ component of the velocity is probed at every 0.02 s starting from 20 s. The complete dataset $\mathbf{X} \in \mathbb{R}^{2647 \times 3001}$ consists of 3001 time samples. More details regarding the simulation setup can be found in the supplementary material.

### 3.4.4 SEA SURFACE TEMPERATURE (SST)

The monthly variation of SST was taken from (Huang et al., 2021; noa). We use the dataset "/Datasets/noaa.oisst.v2.highres/sst.mon.mean.nc" which accounts for monthly values from 1981/09 to 2023/08. We take every $10^{\text{th}}$ sample of latitude and longitude for modeling. Additionally, the dataset was preprocessed to remove anomalous values and instances containing 'NaN'. Since the measurement data already contains noise, we test our method for different training dataset size. The complete dataset $\mathbf{X} \in \mathbb{R}^{6822 \times 503}$ consists of total 503 time samples. The training strategy is same for the rest of the datasets in the original manuscript.

### 3.5 TRAINING STRATEGY

We follow the training strategy as in (Azencot et al., 2020; cKA). We partition the dataset $\mathbf{X}$ into training ($\mathbf{X}_{\text{train}}$) and testing ($\mathbf{X}_{\text{train}}$) set. Guided by one of the core motivations of this paper to analyze the cases with limited data, we analyze three different scenarios for each dataset with three different values of $N_{\text{train}}$. Since we are using the consistent Koopman autoencoder (cKAE) (Azencot et al., 2020) for benchmarking, we perform the hyperparameter search to get best possible performance for cKAE. We then maintain those optimal hyperparameters for cKAE and introduce our proposed loss $\mathcal{L}_{\text{pc}}$. We tune three additional hyperparameters for the proposed pcKAE, namely the weight $\gamma_{\text{pc}}$ corresponding to $\mathcal{L}_{\text{pc}}$, the maximum look-ahead step $\kappa_m$, and the epoch $(e_s)$ for switching to non-zero $\gamma_{\text{pc}}$. Up to $e_s$ we train without enforcing the consistency regularization, and switch to nonzero $\gamma_{\text{pc}}$ from $e_s$ onward. We also observe the effect of noise on model performance for two different noise level, 30 dB and 40 dB (zero mean Gaussian). For different level of noise we tune the weights of different loss components to get best possible accuracy keeping other parameters fixed. For the pendulum, we analyze the effect of noise for all three different values $N_{\text{train}}$, whereas for high-dimensional fluid flow and plasma system we analyze the effect of noise for the largest value of $N_{\text{train}}$. Note that for DAE, we tune the hyperparameters from scratch to obtain best possible performance. The detailed training parameters along with the network architecture are provided in the supplementary material.

The generalizability of the models are assessed by its performance on the test set for long-term predictions. The evaluation of the model's performance is based on the relative 2-norm error denoted by, $\delta_n = \frac{||\hat{\mathbf{x}}_n - \mathbf{x}_n||_2}{||\mathbf{x}_n||_2}$, averaged over prediction region, and variance in the predicted values. The variance is quantified by the 90% confidence interval, defined by the $5^{\text{th}}$ and $95^{\text{th}}$ percentiles of the error, providing a robust summary of the typical error range by excluding the extreme 5% of values at both ends of the distribution. Note that for notational conciseness we provide the width of the 90% interval indicating degree of variance. Note that since we are dealing with time-series, the sequential integrity of the samples in $\mathbf{X}_{\text{train}}$ and $\mathbf{X}_{\text{test}}$ are preserved. First 30 samples from $\mathbf{X}_{\text{test}}$ are fed to the pre-trained models to reconstruct the entire trajectory spanning $\mathbf{X}_{\text{test}}$, which is then compared against the noise-free ground truth to evaluate its performance. The variation in error for multiple inputs also helps assess the robustness of the model.

## 4 RESULTS

We generate the data using the initial condition $\theta_0 = 0.8$, $\dot{\theta} = 0$. One cycle of oscillation is covered approximately by 20 time-steps. We perform the tests for $N_{\text{train}} = 32, 50$ and $90$, with two different noise levels 40 dB and 30 dB for each $N_{\text{train}}$. The relative 2-norm error is calculated over 1900 time-samples in testing set (extrapolation region). The results are summarized in Table 1 and Table 2.

As can be seen in Table 1, our proposed pcKAE has clear advantage over DAE or cKAE for clean data. This advantage is magnified for $N_{\text{train}} = 32$, which covers approximately one and half cycle of oscillation of the pendulum. The advantage tends to diminish with increasing size of training set. Large $N_{\text{train}}$ essentially makes learning with larger look-ahead step $\kappa$ possible for labelled training data, resulting in enhanced stability for long-term predictions for DAE and cKAE. Table 1 also shows

Table 1: Error (%) comparison: DAE, cKAE, and pcKAE on clean pendulum data. The quantities inside parenthesis show the width of 90% confidence interval.

| Method | $N_{\text{train}} = 32$ | $N_{\text{train}} = 50$ | $N_{\text{train}} = 90$ |
|--------|--------|--------|--------|
| DAE | 26.733 (90.11) | 12.119 (43.23) | 11.287 (42.34) |
| cKAE | 9.684 (33.00) | 5.914 (14.52) | 4.578 (21.20) |
| pcKAE | 2.936 (9.20) | 3.131 (5.81) | 3.059 (13.00) |

Table 2: Error (%) comparison for different $N_{\text{train}}$ and noise levels for pendulum data.

| Method | 40 dB SNR | | | 30 dB SNR | | |
|--------|--------|--------|--------|--------|--------|--------|
| | $N_{\text{train}} = 32$ | $N_{\text{train}} = 50$ | $N_{\text{train}} = 90$ | $N_{\text{train}} = 32$ | $N_{\text{train}} = 50$ | $N_{\text{train}} = 90$ |
| DAE | 32.237 (127.65) | 8.364 (22.03) | 9.336 (36.58) | 38.292 (178.52) | 10.533 (35.64) | 11.466 (45.71) |
| cKAE | 9.440 (36.27) | 5.999 (14.19) | 6.588 (41.96) | 17.733 (77.04) | 6.863 (23.02) | 7.684 (32.07) |
| pcKAE | 4.018 (15.13) | 3.832 (8.00) | 3.259 (16.00) | 12.500 (49.70) | 5.494 (15.56) | 6.104 (23.20) |

that the variation of error (values inside parenthesis) is lowest for pcKAE. We see a similar trend with respect to $N_{\text{train}}$ when noise is included. However, interetingly with addition of noise in some cases the relative error actually decreases. This should not be surprising since models tend to overfit with limited data, and adding noise is one of the solutions to avoid overfitting. Overall, pcKAE provides more accurate estimate of the state for long-term prediction.

## 4.1 OSCILLATING ELECTRON BEAM

For the electron beam oscillation, one period approximately comprises of 25 time samples. In order to assess the performance of the Koopman models for limited dataset, we train the models for $N_{\text{train}} = 40, 50$ and 100. For clean data (Table 3), pcKAE performs along the expected line with increase size of training dataset. Since pcKAE seems to have obvious advantage for limited dataset, we only focus on the large dataset ($N_{\text{train}} = 100$) in order to asess the effect of noise. For different noise levels pcKAE fails to provide any significant advantage, indicating that primary advantage of pcKAE lies in handling limited dataset.

Table 3: Results for electron beam oscillation: Error (%) comparison between DAE, cKAE, and pcKAE for clean data (left); for $N_{\text{train}} = 100$ and noisy data (right).

| Method | $N_{\text{train}} = 40$ | $N_{\text{train}} = 50$ | $N_{\text{train}} = 100$ | Method | Clean | 40 dB SNR | 30 dB SNR |
|--------|--------|--------|--------|--------|--------|--------|--------|
| DAE | 7.713 (11.07) | 2.964 (4.32) | 0.770 (0.94) | DAE | 0.770 (0.94) | 1.372 (0.75) | 3.847 (1.68) |
| cKAE | 7.559 (10.19) | 2.468 (2.50) | 0.811 (0.93) | cKAE | 0.811 (0.93) | 1.359 (0.73) | 3.640 (1.05) |
| pcKAE | 2.802 (2.15) | 1.754 (0.85) | 0.802 (0.90) | pcKAE | 0.802 (0.90) | 1.335 (0.65) | 3.560 (0.84) |

## 4.2 FLOW PAST CYLINDER

In case of flow past cylinder, since one cycle approximately contains 30 time samples, we consider $N_{\text{train}} = 40, 50$ and 120. Model performance is tested against different noise levels for $N_{\text{train}} = 120$. The final error is calculated by averaging the 2-norm relative error over 2500 prediction steps beyond the training region. This example is interesting since even for limited data all of the models show good accuracy (Table 4). However, in presence of noise pcKAE outperforms both DAE and cKAE. Note that the absurdly high value for cKAE for 30 dB SNR is most likely an outlier.

## 4.3 SEA SURFACE TEMPERATURE

For the SST data we compare pcKAE only with cKAE for three different training dataset sizes. Interestingly, for SST data the advantage of pcKAE is not very obvious, which points out one of the limitations of pcKAE, the inherent assumption of time-invariance. SST variations are generally not time-invariant due to the impact of multiple time-varying factors, such as atmospheric conditions,

Table 4: Results for flow past cylinder: error (%) comparison between DAE, cKAE, and pcKAE for clean data (left); for $N_{\text{train}} = 120$ and noisy data (right)

| Method | $N_{\text{train}} = 40$ | $N_{\text{train}} = 50$ | $N_{\text{train}} = 120$ | Method | Clean | 40 dB SNR | 30 dB SNR |
|--------|------|------|------|--------|-------|-----------|-----------|
| DAE | 1.405 (2.84) | 0.413 (0.58) | 0.062 (0.06) | DAE | 0.062 (0.06) | 0.635 (0.85) | 3.211 (5.27) |
| cKAE | 2.220 (5.39) | 0.398 (0.59) | 0.064 (0.06) | cKAE | 0.064 (0.06) | 0.555 (0.61) | 9.4887 (21.53) |
| pcKAE | 1.162 (2.477) | 0.375 (0.53) | 0.061 (0.04) | pcKAE | 0.061 (0.04) | 0.484 (0.44) | 1.745 (1.62) |

ocean currents, and longer-term climate patterns. The inherent dynamics of SSTs are influenced by both periodic seasonal changes and non-periodic variations such as El Niño and La Niña events (Ohba & Ueda, 2007), which can inject anomalies in the SST variation.

Table 5: Error (%) comparison between cKAE and pcKAE

| Method | $N_{\text{train}} = 32$ | $N_{\text{train}} = 60$ | $N_{\text{train}} = 120$ |
|--------|------|------|------|
| cKAE | 6.361 (12.67) | 4.444 (7.11) | 4.088 (6.14) |
| pcKAE | 6.310 (12.36) | 4.381 (6.73) | 3.960 (5.84) |

## 5 ABLATION STUDY

Note that the weights $\gamma_{\text{id}}$ (always set to 1) and $\gamma_{\text{fwd}}$ are absolute essential. The ablation study essentially contains one extra case where $\gamma_{\text{pc}} \neq 0$, but $\gamma_{\text{bwd}} = \gamma_{con} = 0$. The ablation studies are done for the extreme cases. We take the best case for DAE and change the $\gamma_{\text{pc}}$ to obtain the optimum result. The ablation study for the oscillating plasma beam is shown here (rest in supplementary material).

Table 6: Ablation study for oscillating electron beam

| | $\gamma_{\text{id}}$ | $\gamma_{\text{fwd}}$ | $\gamma_{\text{bwd}}$ | $\gamma_{\text{con}}$ | $\gamma_{\text{pc}}$ | Average relative error(%) |
|--------|------|------|------|------|------|------|
| $N_{\text{train}} = 40$, clean | 1 | 0.5 | 0 | 0 | 0 | 7.713 |
| | 1 | 0.5 | 0 | 0 | 1e-1 | 3.084 |
| | 1 | 0.5 | 1e-3 | 1e-5 | 0 | 7.559 |
| | 1 | 0.5 | 1e-3 | 1e-5 | 1e-1 | 2.802 |
| $N_{\text{train}} = 100$, clean | 1 | 0.01 | 0 | 0 | 0 | 0.771 |
| | 1 | 0.01 | 0 | 0 | 1e-3 | 0.725 |
| | 1 | 0.05 | 1e-1 | 1e-5 | 0 | 0.811 |
| | 1 | 0.05 | 1e-1 | 1e-5 | 1e-3 | 0.802 |
| $N_{\text{train}} = 100$, 30 dB noise | 1 | 0.01 | 0 | 0 | 0 | 3.847 |
| | 1 | 0.01 | 0 | 0 | 1e-2 | 3.574 |
| | 1 | 0.1 | 1e-1 | 1e-5 | 0 | 3.640 |
| | 1 | 0.1 | 1e-1 | 1e-5 | 1e-2 | 3.560 |

## 6 CONCLUSION

In this paper, we proposed a physics constrained learning (PCL) model named prediction-consistent Koopman autoencoder (pcKAE) which are Koopman autoencoders (KAE) with a prediction consistency test for forecasting time-series data from nonlinear complex systems. The key approach uses the time-invariance property of the Koopman operator for autonomous dynamical systems to enforce prediction consistency in the latent variables. The pcKAE is evaluated on four datasets and compared against other KAE techniques including the most recent cKAE (Azencot et al., 2020). It outperforms the state-of-the-art methods for shorter training length and noisy training data, but have limited advantage when data is abundant or the underlying dynamics is time-variant. The ability of pcKAE to handle limited data can be crucial since simulation of high-dimensional physical system can often be computationally demanding. This method can be extended to non-autonomous control systems with a bilinearly recurrent KAE architecture by using the bilinear structure proposed in (Goswami & Paley, 2022).

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
