# Supplementary Material for "Prediction-Consistent Koopman Autoencoders"

The supplementary material contains the following:

- Details about the simulation setup

- Network architecture and hyperparameters

- Ablation study

## 1   Simulation Setup

In this section we discuss the simulation setup in details for both the oscillating electron beam and flow past cylinder.

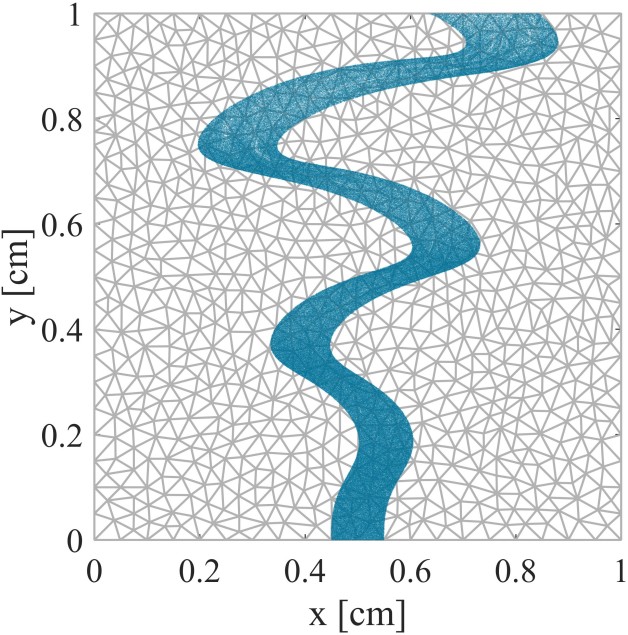

Figure 1: Snapshot of electron beam at $t = 8$ ns.

## 1.1   Oscillating electron beam

A two-dimensional (2-D) electron beam (Fig. 1) is simulated inside a square cavity of dimension 1 cm $\times$ 1 cm using a charge-conserving electromagnetic particle-in-cell (EMPIC) algorithm [1]. The electron beam is propagating along the +ve $y$ direction under the influence of an oscillating transverse magnetic flux. The solution domain is discretized using irregular triangular mesh (grey lines in Fig. 1) with $N_0 = 844$ nodes, $N_1 = 2447$ edges and $N_2 = 1604$ elements (triangles). The blue dots in Fig. 1 represent the superparticles

which are discretized representation of the phase-space of electrons (delta distribution in both position and velocity). The superparticles are essentially the point charges with charge $q_{sp} = r_{sp}q_e$, and mass $m_{sp} = r_{sp}m_e$, where $r_{sp}$ is the superparticle ratio with $q_e$ and $m_e$ respectively representing the charge and mass of an electron. We select $r_{sp} = 5000$. The superparticles are injected from the bottom of the cavity randomly with uniform distribution over the region [0.45 cm, 0.55 cm] with the injection rate of 50 superparticles per time-step. The superparticles are injected along the vertical direction with $v_y = 5 \times 10^6$ m/s. The external oscillating magnetic flux can be represented by $\mathcal{B}_{ext} = \mathcal{B}_0 \sin(2\pi/T_{osc}) \, \hat{z}$ with $\mathcal{B}_0 = 2.5 \times 10^{-2}$ T, and $T_{osc} = 0.8$ ns. The time-step for the EMPIC simulation is taken to be $\Delta t = 0.2$ ps.

## 1.2 Flow past cylinder

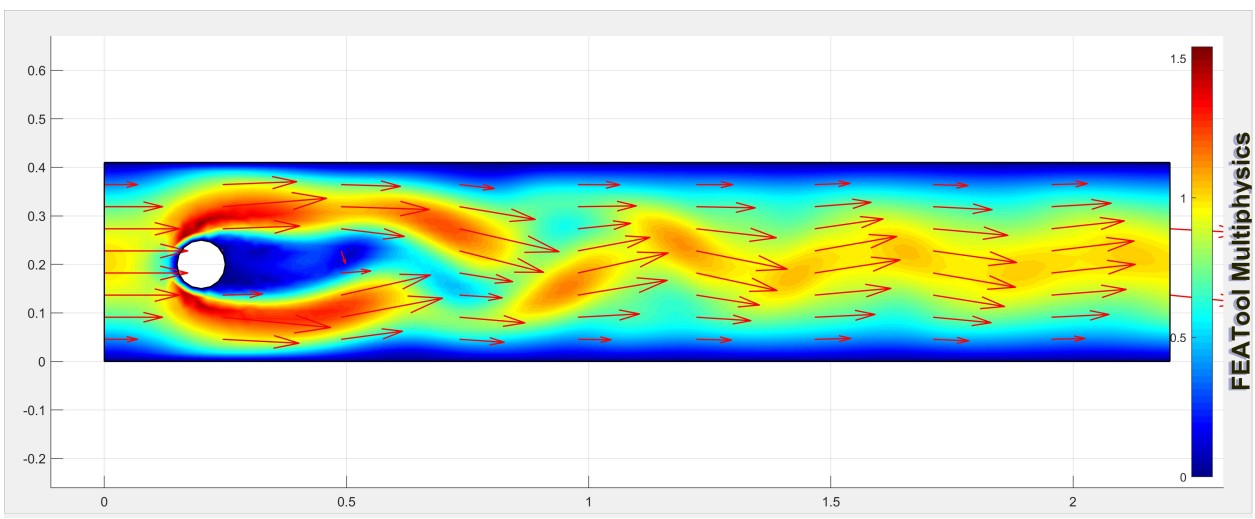

Figure 2: Snapshot of velocity field at $t = 80$ s.

The whole simulation setup is shown in Fig. 2. The 2-D solution domain (2.2 m × 0.41 m) is discretized using irregular triangular mesh with number of nodes $N_0 = 2647$, and number of elements (triangles) $N_2 = 5016$. The cylinder has the diameter of 0.1 m with center of the cylinder located at $(0.2 \text{ m}, 0.2 \text{ m})$. The flow is assumed to be incompressible, and governed by the Navier-Stokes equations with $\mathbf{u}$, $\mathbf{v}$ denoting the horizontal and vertical component of the velocity respectively while $\mathbf{p}$ denotes the pressure field. The density of the fluid is set to $\rho = 1$ Kg/m$^3$, and dynamic viscosity $\mu = 0.001$ Kg/m s. The flow is unsteady with a maximum velocity of 1 m/s and mean velocity $\frac{2}{3}$ of the maximum velocity. The simulation starts with initial conditions $\mathbf{u}_0 = \mathbf{v}_0 = 0$ and $\mathbf{p}_0 = 0$. For the boundary conditions, the leftmost boundary is set as an inlet with a parabolic velocity profile. This is representative of fully developed laminar flow at the inlet. The rightmost boundary is set as an outflow (pressure boundary), where we specify the pressure but do not specify the velocity, allowing the flow to exit naturally based on the internal flow field. All other boundaries are treated as walls with a no-slip condition, which means the fluid velocity at the walls is zero. The simulation runs for a total of 80 seconds, with a time-step size of 0.01 seconds.

## 2  Network architecture and hyperparameters

The training strategy was already discussed in the main body of the paper. In this section we will mainly go into the details of the network architecture and specific values of the hyper-parameters used for training.

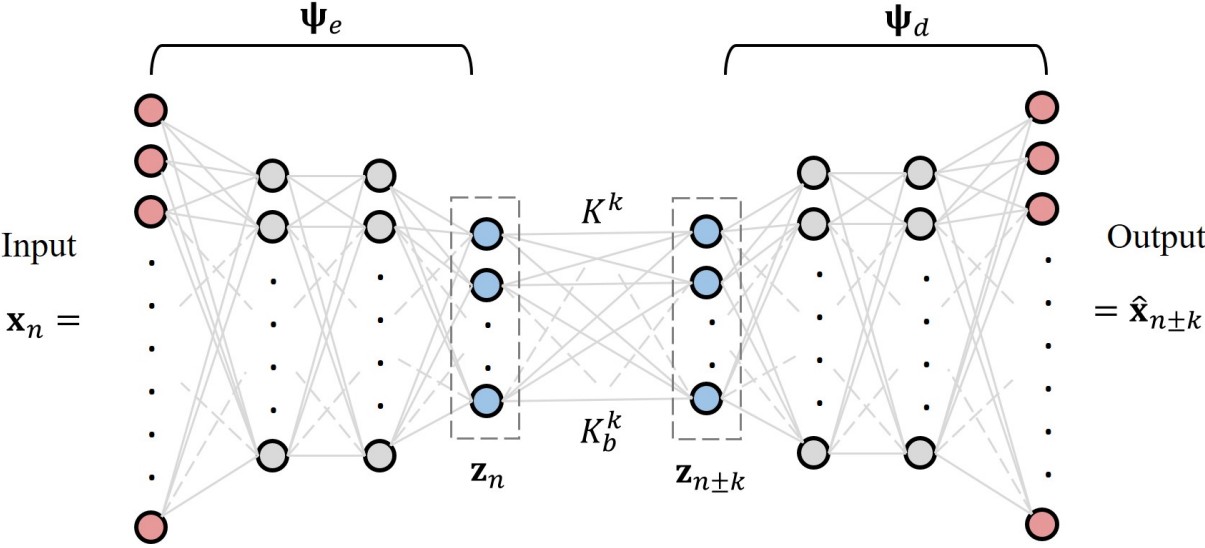

Figure 3: pcKAE architecture where the layers with red nodes indicate the input and output layer in the original state space. The latent space is represented by the bottleneck layer with blue nodes. The hidden layers are shown by the gray nodes.

## 2.1 Network architecture

The network architecture is shown in Fig. 3. Note that the network topology is same as that in [2]. The number of nodes in the input and output layer (red) is indicated by $N_{\text{in}}$ and $N_{\text{out}}$ ($N_{\text{in}} = N_{\text{out}}$). $N_l$ denotes the number of nodes in the bottleneck layer (blue), which represents the approximation of Koopman-invariant latent space. Number of nodes in each hidden layers (two hidden layers for each encoder and decoder) is represented by $N_h$ (gray nodes). We keep $N_h$ same for both the hidden layers and it is tuned in multiple of 8 for the pendulum case and in multiple of 16 for plasma beam and fluid flow case. We use the *tanh* activation function for each layer except the bottleneck layer.

## 2.2 Training details

The crucial hyperparameters to tune are learning rate $(l_r)$, learning rate decay rate $(l_{rd})$, corresponding decay schedule, maximum look-ahead step $\kappa_m$, the weigths of the individual component of the total loss, i.e. $\gamma_{\text{id}}, \gamma_{\text{fwd}}, \gamma_{\text{bwd}}, \gamma_{\text{con}}$ and $\gamma_{\text{pc}}$. Among structural hyperparameters, $N_h$, $N_l$ are crucial. Note that as mentioned in the previous subsection, $N_h$ is varied by varying $\alpha$. We provide values of these hyperparameters for extreme scenarios for each of the test-case. Note that $N_{\text{in}} = N_{\text{out}}$ are not tunable, and depend on the dimension of the input data. The pendulum cases is trained for 600 epochs whereas electron beam and fluid flow cases are trained for 1000 epochs.

### 2.2.1 Oscillating electron beam

Please see Table 1. The learning rate decay schedule for oscillating electron beam is [30, 200, 400, 700], i.e. learning rate is reduced by a factor of 0.5 at epochs 30,70, 200 and 700.

## 2.3 Flow past cylinder

Please see Table 2. The learning rate decay schedule (epochs) is [30, 200, 400, 700] by factor of 0.5 .

Table 1: Training hyperparameters of DAE, cKAE, and pcKAE for oscillating electron beam

| | Method | $N_\mathrm{in}=N_\mathrm{out}$ | $N_h$ | $N_l$ | $l_r$ | $l_{rd}$ | $\kappa_m$ | $\gamma_\mathrm{fwd}$ | $\gamma_\mathrm{bwd}$ | $\gamma_\mathrm{con}$ | $\gamma_{pc}$ | $\kappa_{pm}$ | $e_s$ |
|---|---|---|---|---|---|---|---|---|---|---|---|---|---|
| $N_\mathrm{train}=40$, clean | DAE | 2447 | 256 | 48 | 1e-3 | 0.5 | 10 | 0.5 | - | - | - | - | - |
| | cKAE | 2447 | 256 | 48 | 1e-3 | 0.5 | 10 | 0.5 | 1e-3 | 1e-5 | - | - | - |
| | pcKAE | 2447 | 256 | 48 | 1e-3 | 0.5 | 10 | 0.5 | 1e-3 | 1e-5 | 1e-1 | 45 | 600 |
| $N_\mathrm{train}=100$, clean | DAE | 2447 | 256 | 32 | 1e-3 | 0.5 | 25 | 0.01 | - | - | - | - | - |
| | cKAE | 2447 | 256 | 32 | 1e-3 | 0.5 | 25 | 0.05 | 1e-1 | 1e-5 | - | - | - |
| | pcKAE | 2447 | 256 | 32 | 1e-3 | 0.5 | 25 | 0.05 | 1e-1 | 1e-5 | 1e-3 | 15 | 600 |
| $N_\mathrm{train}=100$, 30 dB noise | DAE | 2447 | 256 | 32 | 1e-3 | 0.5 | 25 | 0.01 | - | - | - | - | - |
| | cKAE | 2447 | 256 | 32 | 1e-3 | 0.5 | 25 | 0.1 | 1e-1 | 1e-5 | - | - | - |
| | pcKAE | 2447 | 256 | 32 | 1e-3 | 0.5 | 25 | 0.05 | 1e-1 | 1e-5 | 1e-2 | 10 | 600 |

Table 2: Training hyperparameters of DAE, cKAE, and pcKAE for flow past cylinder

| | Method | $N_\mathrm{in}=N_\mathrm{out}$ | $N_h$ | $N_l$ | $l_r$ | $l_{rd}$ | $\kappa_m$ | $\gamma_\mathrm{fwd}$ | $\gamma_\mathrm{bwd}$ | $\gamma_\mathrm{con}$ | $\gamma_{pc}$ | $\kappa_{pm}$ | $e_s$ |
|---|---|---|---|---|---|---|---|---|---|---|---|---|---|
| $N_\mathrm{train}=40$, clean | DAE | 2647 | 256 | 64 | 1e-3 | 0.5 | 15 | 1e-2 | - | - | - | - | - |
| | cKAE | 2647 | 256 | 48 | 1e-3 | 0.5 | 15 | 1e-4 | 1e-6 | 1e-4 | - | - | - |
| | pcKAE | 2647 | 256 | 48 | 1e-3 | 0.5 | 10 | 0.5 | 1e-3 | 1e-5 | 1e-3 | 50 | 100 |
| $N_\mathrm{train}=100$, clean | DAE | 2647 | 256 | 64 | 1e-3 | 0.05 | 15 | 0.05 | - | - | - | - | - |
| | cKAE | 2647 | 256 | 48 | 1e-3 | 0.5 | 15 | 0.05 | 1e-4 | 1e-4 | - | - | - |
| | pcKAE | 2647 | 256 | 48 | 1e-3 | 0.5 | 10 | 0.05 | 1e-3 | 1e-5 | 1e-3 | 20 | 400 |
| $N_\mathrm{train}=120$, 30 dB noise | DAE | 2647 | 256 | 64 | 1e-3 | 0.5 | 15 | 2 | - | - | - | - | - |
| | cKAE | 2647 | 256 | 48 | 1e-3 | 0.5 | 15 | 0.5 | 1e-6 | 1e-4 | - | - | - |
| | pcKAE | 2647 | 256 | 48 | 1e-3 | 0.5 | 10 | 0.5 | 1e-3 | 1e-5 | 1e-3 | 20 | 100 |

# 3 Ablation study

Note that the weights $\gamma_\mathrm{id}$ and $\gamma_\mathrm{fwd}$ are absolute essential The ablation study essentially contains one extra case where we do not consider the loss components pertaining to backward dynamics, by chosing $\gamma_\mathrm{bwd} = \gamma_{con} = 0$. The ablation studies are done for the extreme cases. We take the best case for DAE and change the $\gamma_\mathrm{pc}$ to obtain optimum result.

Table 3: Ablation study for undamped pendulum

| | $\gamma_\mathrm{id}$ | $\gamma_\mathrm{fwd}$ | $\gamma_\mathrm{bwd}$ | $\gamma_\mathrm{con}$ | $\gamma_\mathrm{pc}$ | Average relative error(%) |
|---|---|---|---|---|---|---|
| $N_\mathrm{train}=32$, clean | 1 | 1 | 0 | 0 | 0 | 26.73 |
| | 1 | 1 | 0 | 0 | 1 | 02.70 |
| | 1 | 0.5 | 1e-2 | 1e-1 | 0 | 09.684 |
| | 1 | 0.5 | 1e-2 | 1e-1 | 1e-1 | 02.93 |
| $N_\mathrm{train}=90$, clean | 1 | 0.5 | 0 | 0 | 0 | 11.29 |
| | 1 | 0.5 | 0 | 0 | 1e-4 | 7 |
| | 1 | 0.05 | 1e-4 | 1e0 | 0 | 4.58 |
| | 1 | 0.05 | 1e-4 | 1e0 | 1e-2 | 3.06 |
| $N_\mathrm{train}=90$, 30 dB noise | 1 | 0.01 | 0 | 0 | 0 | 11.4657 |
| | 1 | 0.01 | 0 | 0 | 1e-2 | 5.8483 |
| | 1 | 0.01 | 1e-4 | 1e-2 | 0 | 7.6836 |
| | 1 | 0.01 | 1e-4 | 1e-2 | 1e-4 | 6.1037 |

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

Table 4: Ablation study for oscillating electron/plasma beam

|  | $\gamma_{\mathrm{id}}$ | $\gamma_{\mathrm{fwd}}$ | $\gamma_{\mathrm{bwd}}$ | $\gamma_{\mathrm{con}}$ | $\gamma_{\mathrm{pc}}$ | Average relative error(%) |
|---|---|---|---|---|---|---|
| $N_{\mathrm{train}} = 40$, clean | 1 | 0.5 | 0 | 0 | 0 | 7.7134 |
|  | 1 | 0.5 | 0 | 0 | 1e-1 | 3.0843 |
|  | 1 | 0.5 | 1e-3 | 1e-5 | 0 | 7.5587 |
|  | 1 | 0.5 | 1e-3 | 1e-5 | 1e-1 | 2.8021 |
| $N_{\mathrm{train}} = 100$, clean | 1 | 0.01 | 0 | 0 | 0 | 0.7706 |
|  | 1 | 0.01 | 0 | 0 | 1e-3 | 0.7251 |
|  | 1 | 0.05 | 1e-1 | 1e-5 | 0 | 0.8110 |
|  | 1 | 0.05 | 1e-1 | 1e-5 | 1e-3 | 0.8025 |
| $N_{\mathrm{train}} = 100$, 30 dB noise | 1 | 0.01 | 0 | 0 | 0 | 3.8473 |
|  | 1 | 0.01 | 0 | 0 | 1e-2 | 3.5740 |
|  | 1 | 0.1 | 1e-1 | 1e-5 | 0 | 3.6404 |
|  | 1 | 0.1 | 1e-1 | 1e-5 | 1e-2 | 3.5597 |

Table 5: Ablation study for flow past cylinder

|  | $\gamma_{\mathrm{id}}$ | $\gamma_{\mathrm{fwd}}$ | $\gamma_{\mathrm{bwd}}$ | $\gamma_{\mathrm{con}}$ | $\gamma_{\mathrm{pc}}$ | Average relative error(%) |
|---|---|---|---|---|---|---|
| $N_{\mathrm{train}} = 40$, clean | 1 | 1e-2 | 0 | 0 | 0 | 1.4047 |
|  | 1 | 1e-2 | 0 | 0 | 1e-3 | 1.1124 |
|  | 1 | 1e-4 | 1e-6 | 1e-4 | 0 | 2.22 |
|  | 1 | 1e-4 | 1e-6 | 1e-4 | 1e-3 | 1.1615 |
| $N_{\mathrm{train}} = 120$, clean | 1 | 0.05 | 0 | 0 | 0 | 0.062 |
|  | 1 | 0.05 | 0 | 0 | 1e-2 | 0.061 |
|  | 1 | 0.05 | 1e-3 | 1e-2 | 0 | 0.0637 |
|  | 1 | 0.05 | 1e-3 | 1e-2 | 1e-3 | 0.0613 |
| $N_{\mathrm{train}} = 120$, 30 dB noise | 1 | 2 | 0 | 0 | 0 | 3.211 |
|  | 1 | 2 | 0 | 0 | 1e-2 | 1.8703 |
|  | 1 | 0.5 | 1e-4 | 1e-4 | 0 | 9.4887 |
|  | 1 | 0.5 | 1e-4 | 1e-4 | 1e-2 | 1.7453 |