# OpenReview forum: "Prediction-Consistent Koopman Autoencoders"
_ICLR.cc/2024/Conference — ICLR 2024 Conference Withdrawn Submission_

### Official Review · Reviewer_41Am · 2023-10-26

**Soundness:** 3 good
**Presentation:** 2 fair
**Contribution:** 1 poor
**Rating:** 3
**Confidence:** 4

**Summary:**

This paper introduces a method for predicting nonlinear dynamic using Koopman autoencoder neural networks. The authors suggest to augment existing techniques with a prediction constraint that promotes latent states to be linearly related. The method is evaluated on several datasets in comparison to several baseline approaches.

**Strengths:**

The paper is easy to follow; The method is described concisely and effectively; the evaluation section seems to be detailed.

**Weaknesses:**

The main weakness of this paper is the proposal of a loss term that was introduced and used before. Specifically, the paper by Lusch et al. "Deep learning for universal linear embeddings of nonlinear dynamics" discusses the same loss term (see page 4, 'linear dynamics'). Since the paper by Lusch et al., several other works have used a similar term, and it is generally known as one of the loss terms to be used in Koopman-based autoencoder frameworks. Thus, unfortunately, this work is not new from an algorithmic viewpoint.

Further, while the evaluation section is decent, the results are not promising. Specifically, the results in Tables 3, 4, 5 show that there is no statistical significance between cKAE and pcKAE (as measured by the standard deviation). This may explain why cKAE did not consider the additional loss term in their work.

**Questions:**

See above.

---

> ### Author Response · Authors · 2023-11-22
>
> There seems to be a misunderstanding regarding how our $L_{pc}$ is different from $L_{lin}$ as proposed in [1]. Note that $L_{lin}$ as proposed in [1] is limited by the number of samples in training dataset. Let us consider a training dataset $\\{\mathbf{x}_1, \mathbf{x}_2 \\}$ consisting only two time samples of state. Let the corresponding latent space representation be $\\{\mathbf{z}_1, \mathbf{z}_2 \\}$ where $\mathbf{z}_i\approx \Psi_e(\mathbf{x}_i)$. Now based on this,
>
> Linear loss in [1]: $L_{lin} = || \mathbf{z}\_2 - K\mathbf{z}\_1 ||$
>
> Our proposed loss: $L_{pc} = \sum_{\kappa=1}^{\kappa_m} || K^\kappa \mathbf{z}_2 - K^{\kappa + 1} \mathbf{z}_1 ||$, where $\kappa_m$ is a hyperparameter.
>
> The main difference being our proposed $L_{pc}$ is not limited by the number of samples in training dataset.
>
> [1] B. Lusch, J. N. Kutz, S. L. Brunton, Deep learning for universal linear embeddings of nonlinear dynamics, Nature communications 9 (1) (2018) 4950.

---

> > ### Comment · Reviewer_41Am · 2023-11-22
> >
> > I would like to thank the authors for their response. However, the linear loss in [1] is controlled by a hyperparameter for the number of steps, and while the authors of [1] may not consider the case you mention, it is covered by their approach. Moreover, the authors of the current submission should have justified more thoroughly the importance of this hyperparameter, its effect on forecasting, and corresponding analysis. All of these aspect are unfortunately missing in this submission.

---

> ### Author Response · Authors · 2023-11-22
> **Comment on Statistical Significance of the Results**
>
> Regarding author's comment on statistical significance of the results, we would like to point out that for limited data set, pcKAE outperforms cKAE significantly for Table 2, Table 3 and Table 4. Note that since advantage of pcKAE is obvious for limited data, we have tested noisy data only for large training set for Table 3 and Table 4. We have intentionally included Table 5 to show possible drawbacks of pcKAE. Since the proposed loss term in pcKAE exploits time-invariance nature of the dynamical system, for time-varying system it might not be that advantageous. As we have mentioned in the manuscript, "The inherent dynamics of SSTs are influenced by both periodic seasonal changes and non-periodic variations such as El Niño and La Niña events (Ohba & Ueda, 2007), which can inject anomalies in the SST variation."

---

### Official Review · Reviewer_RvL6 · 2023-10-29

**Soundness:** 4 excellent
**Presentation:** 3 good
**Contribution:** 3 good
**Rating:** 8
**Confidence:** 2

**Summary:**

The paper proposes a physics constrained autoencoder for time-series data forecasting. It builds on existing work done on Koopman Autoencoder and it's subsequent improvement consistent Koopman Autoencoder to propose prediction consistent KAE. The main idea of the paper is to use the time invariance property of the Koopman operator to enforce prediction consistency in the latent space. The final result is an added regularization term on top of the loss used in its predecessor cKAE.

**Strengths:**

Even though the ultimate contribution of the paper boils down to proposing a regularization term, the regularization itself is well motivated from a theoretical standpoint and is very clearly backed up by the improvements shown in the experiments. So, while the model may not be completely original it is definitely a significant improvement over its predecessors. The paper is also clearly presented and the experiments section is very thoroughly and fairly done. It is nice to see the confidence intervals and not just mean results being presented, and also full details of hyperparameter optimization being presented and fairly held the same across competing methods. The improvements in the results seem drastic and very significant!

**Weaknesses:**

I am not an expert in this field and had a hard time finding any weaknesses in the paper. But when I see a loss function of the form presented in Eq 12 it does make me wonder about both the data and time cost of grid search on those hyperparameters.

**Questions:**

I'd appreciate author's comment on the question raised in weakness section.

From a visual perspective, it might also be worth adding a figure showing the predictive performance.

---

> ### Author Response · Authors · 2023-11-22
>
> We thank the reviewer for valuable comments. The issue the reviewer raised regarding multiple loss functions is a relevant one, but not unique to pcKAE, rather present in all KAEs [1,2]. This issue is common in multi-objective learning where one has to deal with multiple loss functions. The weights are essentially the hyperparameters, and needs tuning through grid search or random search. In this case, our method is not any different from [1,2]. Although there are no strict rules for choosing the weights, typically the weights corresponding to identity and forward loss i.e. $\gamma_{id}$ and $\gamma_{fwd}$ will have the higher values compared to the rest. This is because $\gamma_{id}$ is used to train the encoding and decoding of the state using labelled data, ensuring the basic autoencoder operation. Since we are mostly interested in forward prediction, $\gamma_{fwd}$ also plays an important role.
>
> We agree that it is worth adding figures for visual perspective. However, for the initial draft we couldn't add the figures due to the page limit. We thank the reviewer for the suggestion.
>
> [1] B. Lusch, J. N. Kutz, S. L. Brunton, Deep learning for universal linear embeddings of nonlinear dynamics, Nature communications 9 (1) (2018) 4950.
>
> [2] O. Azencot, N. B. Erichson, V. Lin, M. Mahoney, Forecasting sequential data using consistent koopman autoencoders, in: International Conference on Machine Learning, PMLR, 2020, pp. 475–485.

---

### Official Review · Reviewer_NiJE · 2023-11-05

**Soundness:** 2 fair
**Presentation:** 2 fair
**Contribution:** 1 poor
**Rating:** 1
**Confidence:** 5

**Summary:**

This manuscript introduced a prediction-consistent Koopman autoencoder (pcKAE) for predicting the behavior of dynamical systems. The authors state that by introducing the prediction consistency loss which satisfies the mathematical constraint, their pcKAE model leads to higher expressivity and generalizability. The authors provided some interesting findings on the dynamical system learning with the help of Koopman theory. The results seem to support the authors’ conclusion.

**Strengths:**

++ The incorporation of the prediction consistency loss improves the long-term predictability of cKAE.

++ The paper is easy to read and understand.

**Weaknesses:**

-- The novelty of the paper is very limited. Adding the prediction consistency loss to the training process as a regularizer is not new, which has been used in many other similar models.

-- The experiments considered to demonstrate the capability of the model are rather simple. The authors should test the method on other complex systems, e.g., 2D/3D GS reaction-diffusion equations, 2D homogeneous isotropic turbulence at Re > 1000, etc.

**Questions:**

1. The authors should provide more creative thinking on the network structure or put forward some further theoretical analyses of the method instead of just doing some obvious mathematical derivation. For example, the authors may find it helpful to improve the latent space learning by adding Fourier transformation, e.g., introduced in [1], making the operator learning to focus on the frequency quantities of the dynamics. Such an improvement may make their work distinguish from the simple incremental one.

2. The author should design the experiments more carefully to prove the advantages of the mode. In particular, the experiments considered to demonstrate the capability of the model are rather simple. The authors should test the method on other complex systems, e.g., 2D/3D GS reaction-diffusion equations, 2D homogeneous isotropic turbulence at Re > 1000, etc.

3. The author should pay more attention on the writing. On page 4, the formulation of the forward loss, I guess, represents the $k$-step prediction of $\hat{\mathbf{x}}_{n+k}$. For different $n$, the symbol should have different meaning. Such a notation may lead to ambiguity. A similar issue exists on the formulation the backward loss of the same page. On page 7, the beginning of subsection 3.5, the authors used the same symbol to represent the train and test data by mistake. It's necessary to use more accurate notations in the manuscript for the sake of preciseness of scientific writing.

[1] Xiong, Wei et al. “Koopman neural operator as a mesh-free solver of non-linear partial differential equations.” ArXiv abs/2301.10022 (2023)

---

> ### Author Response · Authors · 2023-11-22
> **Weakness 1,2 and Qustions 1,2 and 3**
>
> Weakness:
>
> W1. To the best of our knowledge we have not come across any work implementing prediction consistency for time-series prediction. If the reviewer can point out the reference, we would be grateful.
>
> W2. While we agree that turbulence at Re$> 1000$ is more challenging to model, the vortex shedding with Re $=100$, have been used in [2] as a benchmark problem.  Fluid flow past cylinder with vortex shredding have been used as a benchmark problem in data-driven modeling of dynamical system [1,3,4]. In fact, [2] mentions "This model has been a benchmark in fluid dynamics for decades...". The pendulum example is also used extensively for benchmarking data-driven methods [1,5,6,7,8]. Authors in [1] mentions "The nonlinear (undamped) pendulum (Hirsch et al., 1974) is a classic textbook example for dynamical systems, which is also used for benchmarking deep models...". So we respectfully disagree with the reviewer's remark regarding benchmark dataset. Ideally, no number of experiments is enough. But we have tried to cover wide range of physical systems from benchmark test-cases like pendulum, flow-past cylinder to plasma and sea-surface temperature data.
>
> Questions:
>
> Q1. While the idea presented in this work is rather intuitive and "simple", we do not think that as a disadvantage. Rather we think its simplicity yet effectiveness, makes it more attractive. With this simple prediction consistency term we are able to outperform state-of-the art KAE models for limited and noisy training data. The mathematical derivations might not be very complicated, but required to establish a theoretical backing of our proposed work. The reviewers suggestion regarding learning in Fourier space is interesting, and we will look into it. We thank the reviewer for the suggestion.
>
> Q2. This is addressed in W2.
>
> Q3. With due respect, we could not quite comprehend the concern with the notation. Different $n$ denotes different time-samples in the training dataset for which we want to calculate the forward/backward loss. For example, if our training set is $\{\mathbf{x}_1,\mathbf{x}_2,\mathbf{x}_3,\mathbf{x}_4\}$, and we choose $k=2$, then we for forward loss we will take $n=1,2~(M=2)$. We will apply the learned koopman operator twice on $\mathbf{x}_1$ and $\mathbf{x}_2$ to get $\hat{\mathbf{x}}_3$ and $\hat{\mathbf{x}}_4$ respectively. We will then compare $\hat{\mathbf{x}}_3$ with ${\mathbf{x}}_3$ and $\hat{\mathbf{x}}_4$ with ${\mathbf{x}}_4$.
>
> References:
>
> [1] B. Lusch, J. N. Kutz, S. L. Brunton, Deep learning for universal linear embeddings of nonlinear dynamics, Nature communications 9 (1) (2018) 4950.
>
> [2] O. Azencot, N. B. Erichson, V. Lin, M. Mahoney, Forecasting sequential data using consistent koopman autoencoders, in: International Conference on Machine Learning, PMLR, 2020, pp. 475–485.
>
> [3] S. L. Brunton, J. L. Proctor, J. N. Kutz, Discovering governing equations from data by sparse identification of nonlinear dynamical systems, Proceedings of the national academy of sciences 113 (15) (2016) 3932–3937.
>
> [4] S. Bagheri, Koopman-mode decomposition of the cylinder wake, Journal of Fluid Mechanics 726 (2013) 596–623.
>
> [5] S. Greydanus, M. Dzamba, J. Yosinski, Hamiltonian neural networks, Advances in neural information processing systems 32 (2019).
>
> [6] Z. Chen, J. Zhang, M. Arjovsky, L. Bottou, Symplectic recurrent neural networks, in: International Conference on Learning Representations, 2020.
>
> [7] O. Bounou, J. Ponce, J. Carpentier, Online learning and control of dynamical systems from sensory input, in: NeurIPS 2021-Thirty-fifth Conference on Neural Information Processing Systems Year, 2021.
>
> [8] B. Azari, D. Erdogmus, Equivariant deep dynamical model for motion prediction, in: International Conference on Artificial Intelligence and Statistics, PMLR, 2022, pp. 11655–11668.

---

> > ### Comment · Reviewer_NiJE · 2023-11-22
> > **Response to the authors' rebuttal**
> >
> > I don't think my questions and comments are well addressed. In particular, I don't buy the authors' argument "no number of experiments is enough". The examples considered in the paper are too simple, which cannot support the conclusion that the proposed method is powerful in modeling complex spatiotemporal dynamics. A proper selection of test cases covering different levels of difficulty is essential. Hence, I maintain my original rating of rejecting the paper for solid reasons.

---

### Official Review · Reviewer_dPS3 · 2023-11-07

**Soundness:** 1 poor
**Presentation:** 1 poor
**Contribution:** 1 poor
**Rating:** 1
**Confidence:** 5

**Summary:**

This paper proposes the predication-consistent Koopman autoencoder (pcKAE), which introduces a consistency regularization term that enforces consistency among predictions at different time-steps. It is capable of accurate long-term predictions with limited and noisy training data. The paper also presents an analytical justrification for consistency regularization using the Koopman spectral theory. The paper performs comparative experiments on 4 classic nonlinear systems or datasets to show its performance.

**Strengths:**

The paper proposes a new type of regularisation term, which contributes to better Koopman autoencoder for long-term prediction.

**Weaknesses:**

1. The statements or descriptions of the conclusions or technical points sometimes are fairly casual or inaccurate. We strongly recommend the authors to dive into the Koopman theory and address them carefully and rigorously. Like, "learn a reduced-order feature space exhibiting simpler linear dynamics", "Koopman operator maps *between* infinite-dimensional function spaces", "the dynamics can be linearly approximated", "by the finite-dimensional Koopman operator", etc. It seems the authors use the LTI system perspective to understand the linear property of Koopman operator, the dynamics of functionals (ie. observables) is linear, etc.
2. Theorem 1, as the main result in theory, is not satisfactorily rigorous in math, which even involves mistakes (e.g., what is exactly the G? It is not matched with what claimed in its proof). If the latent space is just G, according to your math description, it is just a set of N_l functionals/elements?? And, the theorem does not carefully addresses the conditions of the flow or trajectory, x_t. The proof seems using the coordinates of the operator to show something, please address rigorous in math.
3. The contribution of pcKAE may not be promising. As far as we can see, the paper contributes by introducing the so-called "prediction-consistency" regularisation term, which however is straightforward for enhancing the k-step predictiability.
4. The authors claimed the performance of pcKAE for long-term and high-dimensional prediction from noisy data. For long-term prediction, we assume pcKAE achieves it by increase the time-span in your proposed regularisation term, neglecting its practicability. For high-dimensional point, the authors seem misunderstanding this concept. Our nonlinear system eq.(1) is the state-space model without output equation, where the state is usually multivariate. High-dimensional (statistical) modeling or learning usually refers to such a task that the dimension of problem is so high that the data is deficit. Moreover, as indicated by the autoencoder (AE) word in pcKAE, is the dimension of latent layer N_l even smaller than the state dimension N_d? If so, yours actually deal with a very special case, where the dimension of the Koopman invariant space (that is large enough to model the given flow) is finite and rather small (smaller than the state dim). Actually the Koopman approach implicitly acqures the finite-dim lifted space is high enough, since it is used to approximate an infinite-dim space of functionals. Well, this is not argument for pcKAE only, it is for all KAE structures. The last point for noisy data: as we know for the Koopman setup, we are building the Koopman-based identification framework for state-space equation without process noise, as eq.(1); you have to be careful when addressing any properties or performance for noisy-data performance.

5. The comparative study is not enough. As the literature review has addressed, the paper has to show that pcKAE can really help to improve the performance of nonlinear system identification in the Koopman perspective.  There are many Koopman-based neural network models for time-series prediction. The only improvement over KAE showed in experiments may not convince readers of the values of pcKAE  for nonlinear system modeling.

**Questions:**

1. There seems mistakes in eq.(7), where there are matrix dimension-matching issues.
2. The propose loss function eq.(12) for pcKAE consists of so many regularisation terms, where how these regularisation parameters can be tuned in practice. Are the performance sensitive to the choice of these parameters?
3. What do you mean by "consistency" in your proposed regularisation term? It seems it is nothing related to the "consistency" in statistics or any well-known concept.

---

> ### Author Response · Authors · 2023-11-22
> **Point 1,2 and 3**
>
> Due to the character limit, we will respond via multiple comments. This comment  addresses point 1, 2 and 3.
>
> 1. We respectfully point out that we could not quite comprehend the issues with the mentioned statements. Let us address them one by one:
>
> a) "learn a reduced-order feature space exhibiting simpler linear dynamics" : The idea of autoencoders is to compress the high-dimensional data to a low dimensional ("reduced-order) feature space, where the dynamics is linear and ``simpler" because of ease of analysis, prediction and control compared to non-linear counterpart.
>
>  b) "Koopman operator maps between infinite-dimensional function spaces": By nature, Koopman operator operates on functionals which are infinite dimensional.
>
>  c) "the dynamics can be linearly approximated": The idea of using neural network is to approximate the finite-dimensional Koopman invariant subspace which exhibits linear dynamics.
>
>  d) "by the finite-dimensional Koopman operator" : We agree that the sentence could be restructured as ``by the finite-dimensional approximation of Koopman operator", for better understanding. But it should be clear from the previous sentence which states "Machine learning techniques utilize a finite-dimensional approximation of the Koopman operator by assuming the existence of a finite-dimensional Koopman invariant function space.".
>
>  e)  "It seems the authors use the LTI system perspective to understand the linear property of Koopman operator, the dynamics of functionals (ie. observables) is linear, etc.": Under the Koopman invariance assumption, the latent state dynamics is LTI by the virtue of time-homogeneity of the original nonlinear system. If we can find a finite-dimensional Koopman invariant subspace, the restriction of the infinite-dimensional Koopman operator is indeed a finite-dimensional linear operator. All the major works [1,2] in Koopman based forecasting inherently assumes the time-invariance of the dynamics, since the prediction loss for training is evaluated at multiple time-steps for the same operator. There is also plethora of literature on DMD and EDMD [4,5] which essentially uses the LTI system perspective to understand the linear property of finite-dimensional approximation of the Koopman operator.
>
>
>
> 2. We agree that there is a slight abuse of notation while writing $\mathcal{G}$. In the theorem $\mathcal{G}$ is mistakenly written as just the enumeration of the dictionary, i.e., the set of $N_l$ functionals. It should be the latent functional space (the one that should be Koopman invariant) is $\operatorname{span} \\{ \psi_1(\cdot)\ldots,\psi_{N_l}(\cdot)\\}$. The theorem should be read: Let $\mathbf{\Psi}(\cdot)=[\psi_1(\cdot),\ldots,\psi_{N_l}(\cdot)]^T \in \mathcal{F}^{N_l}$ denotes a vector valued function comprised with scalar functions $\psi_i(\cdot)\in \mathcal{F}$. The latent function space $\mathcal{G} = \operatorname{span}\\{\psi_1(\cdot)\ldots,\psi_{N_l}(\cdot)\\}$ forms a Koopman invariant subspace with respect to the system dynamics (1) if and only if there exists $K\in \mathbb{R}^{N_l\times N_l}$ such that $\mathbf{\Psi}_e (\mathbf{x}\_{n+\kappa})=K^{\kappa} \mathbf{\Psi}_e(\mathbf{x}\_{n})$ for all $n \geq 0$ and $ \kappa\geq 1$.
>
> In the proof, $\operatorname{span}\mathcal{G}$ should be replaced by $\mathcal{G}$ which itself is a span of observables. The authors apologize for this inadvertent mistake and corrects it in the revised version.
>
> The authors are unable to understand what the reviewer meant by ``coordinates of the operator", a clarification will be greatly appreciated.
>
> 3. We respectfully disagree with the reviewer regarding promise of this work. We believe that promise of the work should be judged by the results. We have tested our method for benchmark problems (pendulum, flow-past cylinder) problems, and have shown improvement (in some cases drastic) in long-term prediction accuracy in presence of noise of limited data. The fact that the a simple and intuitive idea can make such significant improvement, makes this work promising.
>
> References:
>
> [1] B. Lusch, J. N. Kutz, S. L. Brunton, Deep learning for universal linear embeddings of nonlinear dynamics, Nature communications 9 (1) (2018) 4950.
>
> [2] O. Azencot, N. B. Erichson, V. Lin, M. Mahoney, Forecasting sequential data using consistent koopman autoencoders, in: International Conference on Machine Learning, PMLR, 2020, pp. 475–485.
>
> [3] J. N. Kutz, S. L. Brunton, B. W. Brunton, J. L. Proctor, Dynamic mode decomposition: data-driven modeling of complex systems, SIAM, 2016.
>
> [4] M. O. Williams, I. G. Kevrekidis, and C. W. Rowley, "A data-driven approximation of the Koopman operator: Extending dynamic mode decomposition," Journal of Nonlinear Science, vol. 25, no. 6, pp. 1307-1346, 2015.

---

> ### Author Response · Authors · 2023-11-22
> **Point 4, 5**
>
> 4. We thank the reviewer for pointing out few interesting points. We have addressed them pointwise and clarified few :
>
>     a) "For long-term prediction, we assume pcKAE achieves it by increase the time-span in your proposed regularisation term, neglecting its practicability." : The look-ahead steps or as the reviewer calls it ``time-span" is a crucial hyperparameter for training Koopman autoencoders (KAE) for accurate long-term prediction. However, this accuracy comes at the cost of training time. This is not only drawback of KAE, but any long-term forecasting model which uses multiple look-ahead steps. Now with limited data we do not have the luxury of a trade-off between training time and accuracy because we don't have the option available for large look-ahead steps. So, with limited data with state-of-the-art KAE fails to produce accurate long-term predictions. But our proposed prediction consistency term enables KAE to look ahead multiple steps. Obviously it comes at the increased training-time, but without this loss the predictions would have been anyway useless.
>
>      b) "For high-dimensional point, the authors seem misunderstanding this concept.": The reviewer most probably misunderstood the nature of ``high-dimensionality" we are discussing in this paper. Note that here we are concerned about the high-dimensional physical systems such as fluid, plasma, weather systems etc. Since we are mostly dealing with high-fidelity spatio-temporal simulations (the test cases involving fluid and plasma), the high-dimensionality arises from the discretization of the partial differential equation (PDE), to be very specific, the discretization of the spatial domain via large number of mesh elements (nodes, edges etc.). For experimental data (SST), it corresponds to different spatial(sensor) locations in sea surface. It is different from the high-dimensional statistical modeling since in spatio-temporal physical systems, there is often some spatial dependencies, typically resulting in an underlying low-dimensional structure.
>
>      Since we are searching for a `reduced"-order model, we intentionally keep the number of nodes in bottleneck layer less than the number of input nodes. We mention this in the abstract as well. In this respect the goal of the paper aligns more with [2] compared to [1]. We understand that reviewer is concerned in general about the nature of KAE itself. The underlying assumption for applying KAE is the existence of a low-dimensional dynamics. Some other works where autoencoders are used for reduced-orde modeling of high-dimensional PDEs are [3,4,5]. A typical example of such Koopman-operator based reduced-order modeling is DMD [7] and EDMD [8]. Note that KAEs are just the neural network version of EDMD.
>
>      c) "The last point for noisy data ... for noisy-data performance.": We thank the reviewer for mentioning this important point. We agree that we have to be careful and specific while addressing our model's noise handling property. One key factor which most probably helps mitigate the zero-mean Gaussian noise is some kind of ``averaging" effect when we enforce prediction consistency among different predictions. For studying the effect of noise, we have followed similar approach as in [2].
>
> 5. With due respect, we did not understand the concern here since we have shown improvement over the state-of-the-art consistent KAE [2] (benchmark) for limited and noisy data for benchmark test-cases like pendulum and flow-past cylinder. Furthermore, [2] assumes existence of a backward dynamics, whereas our pcKAE does not make such assumption. We agree that use of "prediction-consistency" is not specific or limited to Koopman perspective, but we see it as a strength since it can be extended to any time-invariant system.
>
> [1] B. Lusch, J. N. Kutz, S. L. Brunton, Deep learning for universal linear embeddings of nonlinear dynamics, Nature communications 9 (1) (2018) 4950.
>
> [2] O. Azencot, N. B. Erichson, V. Lin, M. Mahoney, Forecasting sequential data using consistent koopman autoencoders, in: International Conference on Machine Learning, PMLR, 2020, pp. 475–485.
>
> [3] L. Agostini, Exploration and prediction of fluid dynamical systems using auto-encoder technology, Physics of Fluids 32 (6) (2020).
>
> [4] N. B. Erichson, M. Muehlebach, M. W. Mahoney, Physics-informed autoencoders for lyapunov-stable fluid flow prediction, arXiv preprint arXiv:1905.10866 (2019).
>
> [5] I. Nayak, M. Kumar, F. L. Teixeira, Koopman autoencoders for reduced-order modeling of kinetic plasmas, Advances in Electromagnetics Empowered by Artificial Intelligence and Deep Learning (2023) 515–542.
>
> [7] P. J. Schmid, Dynamic mode decomposition and its variants, Annual Review of Fluid Mechanics 54 (2022) 225–254.
>
> [8] M. O. Williams, I. G. Kevrekidis, and C. W. Rowley, "A data-driven approximation of the Koopman operator: Extending dynamic mode decomposition," Journal of Nonlinear Science, vol. 25, no. 6, pp. 1307-1346, 2015.

---

> ### Author Response · Authors · 2023-11-22
> **Questions 1,2 and 3**
>
> Q1. We thank the reviewer for pointing it out. There is a typo where $*$ and $i$ should be interchanged in the subscript of $K$ in the second term.  We will correct it in the revised manuscript.
>
> Q2. The issue the reviewer raised is a relevant one, but not unique to pcKAE, rather present in all KAEs [1,2]. This issue is common in multi-objective learning where one has to deal with multiple loss functions. The weights are essentially the hyperparameters, and needs tuning through grid search or random search. In this case, our method is not any different from [1,2]. Although there are no strict rules for choosing the weights, typically the weights corresponding to identity and forward loss i.e. $\gamma_{id}$ and $\gamma_{fwd}$ will have the higher values compared to the rest. This is because $\gamma_{id}$ is used to train the encoding and decoding of the state using labelled data, ensuring the basic autoencoder operation. Since we are mostly interested in forward prediction, $\gamma_{fwd}$ also plays an important role.
>
> Q3. In statistics, consistency usually refers to a property of an estimator where the estimates it produces converge to the true parameter value as the sample size increases. But in our case consistency means similarity between two predictions. So, in our case, higher consistency refers to the reduction of variance among different estimates.
>
> [1] B. Lusch, J. N. Kutz, S. L. Brunton, Deep learning for universal linear embeddings of nonlinear dynamics, Nature communications 9 (1) (2018) 4950.
>
> [2] O. Azencot, N. B. Erichson, V. Lin, M. Mahoney, Forecasting sequential data using consistent koopman autoencoders, in: International Conference on Machine Learning, PMLR, 2020, pp. 475–485.

---

> ### Comment · Reviewer_dPS3 · 2023-11-23
>
> We thank the authors for their detailed responses and further clarifications!
>
> Let me summarize our comments. Before that we would like to point out that we are familiar with both Koopman theory, including Koopman operator (math), Koopman-based system identification (control engineering), Koopman-based nonlinear system modeling (complex systems in physics), and deep learning. Almost all papers you listed had been read by reviewers. So there is no knowledge gap or misunderstanding between our backgrounds.
>
> Our arguments on weakness are mainly from the following two sides:
>
> - (Koopman theory/methods) Your phrases/descriptions/clarifications on technical points and math are fairly careless. Your article hasn't been well-cooked. We really cannot accept your math "typos" appearing even in major problem formulation, theorems ,etc. When you address something on Koopman, (what is "finite-dim", "infinite-dim", can or cannot be approximated, when approximated do we assume finite point-spectra or existence of invariant subspace, etc.,) all these details have to be carefully handled.  Nonlinear system modeling is complicated, and you have to carefully draw a boundary on what your method can solve.
>
> - (deep learning) We are sorry to say that your contribution is minor. Proposing a regularization term additionally, without throughout analysis/argument in theory or convincing experiments, is not qualified for publishing ICLR. From your article, we might predicate you may either an early researcher or  a researcher that is new to either of deep learning or Koopman nonlinear modeling. We strongly suggest the author to deepen your understanding of Koopman theory and deep learning, think about your method strictly in math, and, if theoretical analysis is difficult (mostly in NN), test it throughout in experiments. We understand that there are early papers on Koopman or Koopman-based NN getting published with a few toy experiments, while they are valued by their pioneered work.
>
> Thus, it is unfortunate to say that your current article is not qualified for ICLR. Note that this does not imply your research topic is not qualified. The evaluation is drawn purely for your paper quality and your current outputs presented.

---

> > ### Author Response · Authors · 2023-11-23
> >
> > Thank you for your detailed review and feedback on our manuscript. Although we have some disagreements, we sincerely appreciate your expertise and the opportunity to enhance our work.
> >
> > We acknowledge that certain sections need greater mathematical precision and clarity, but this does not necessarily negate the analytical justification we have provided for using the proposed loss term. We agree with the reviewer that we did not delve deeply into the Koopman spectral theory and intricacies of nonlinear dynamics, given the applied nature of the paper. Including a detailed and careful discussion about the existence of a Koopman invariant subspace, the nature of Koopman spectra etc. could indeed further strengthen our work. However, for this paper, our primary focus has been on 1) providing analytical justification for using the proposed loss term, 2) conducting experiments for several test cases, and 3) fitting everything within the 8-page limit.
> >
> > While we understand the reviewer's concern regarding the novelty being limited to a new regularizing term, we still believe that the model's performance should not be overlooked. It is important to note that [1] (our benchmark) is not an early KAE work, and its novelty also lies in the introduction of new regularizing terms. Therefore, we do not fully agree with the reviewer's comment about the novelty of our paper. Although early pioneering Koopman NN papers had relatively simple test cases, [1] is not an early paper, and it also used similar test cases to ours. In fact, we included an additional test case of nonlinear plasma dynamics in our study. We understand that incorporating more challenging cases can make the paper stronger, but including those benchmark cases was necessary. We find it unfortunate that the reviewer formed the impression that our team consists of early career researchers, given that we have well-published works on Koopman-based modeling. Regardless, we heartily thank the reviewer for their time and valuable comments. We will strive to incorporate necessary changes to strengthen our paper.
> >
> > [1] O. Azencot, N. B. Erichson, V. Lin, M. Mahoney, Forecasting sequential data using consistent Koopman autoencoders, in: International Conference on Machine Learning, PMLR, 2020, pp. 475–485.